# Safeguarding genome integrity during gene-editing therapy in a mouse model of age-related macular degeneration

Jianhang Yin[1,5], Kailun Fang[2,5], Yanxia Gao[2,5], Liqiong Ou[1], Shaopeng Yuan[1], Changchang Xin[1], Weiwei Wu[2], Wei-wei Wu[2], Jiaxu Hong [3,4], Hui Yang [2] ✉ & Jiazhi Hu [1] ✉

Ensuring genome safety during gene editing is crucial for clinical translation of the high-efficient CRISPR-Cas9 toolbox. Therefore, the undesired events including chromosomal translocations, vector integrations, and large deletions arising during therapeutic gene editing remain to be adequately addressed or tackled in vivo. Here, we apply CRISPR-Cas9TX in comparison to CRISPR-Cas9 to target *Vegfa* for the treatment of age-related macular degeneration (AMD) disease in a mouse model. AAV delivery of both CRISPR-Cas9 and CRISPR-Cas9TX can efficiently inhibit laser-induced neovascularization. Importantly, Cas9TX almost eliminates chromosomal translocations that occur at a frequency of approximately 1% in Cas9-edited mouse retinal cells. Strikingly, the widely observed AAV integration at the target *Vegfa* site is also greatly reduced from nearly 50% of edited events to the background level during Cas9TX editing. Our findings reveal that chromosomal structural variations routinely occur during in vivo genome editing and highlight Cas9TX as a superior form of Cas9 for in vivo gene disruption.

Adeno-associated virus-mediated (AAV) Cas9 targeting shows great potential for curing complex diseases in vivo[1–5]. For example, the treatment for severe retinal dystrophy in Leber congenital amaurosis type 10 (LCA10) is in phase 2 in clinics (NCT03872479). However, the usage of AAV-delivered CRISPR-Cas9 for in vivo treatment still faces substantial hurdles in terms of safety, including potential unintended alterations such as chromosomal structural variations and AAV integrations. Chromosomal structural variations are widely observed at a frequency of up to 10% in cultured cells[6–11], but a comprehensive assessment of editing safety for terminally differentiated cells in vivo is still lacking. Besides, the integration of exogenous DNA including AAV and plasmid vectors into the genome at a frequency of up to 47% has

recently been reported during in vivo CRISPR-Cas9 editing[12–14], and the solution for avoiding vector integration is also lacking.

With the safety concern, the unwanted editing byproducts may induce unexpected consequences during genome editing. In this regard, chromosomal translocations have been linked to tumorigenesis, including but not limited to leukemia, prostate cancer, breast cancer, and Ewing's sarcoma[15–18]. The integrated DNA elements might affect the gene expression in the neighbor region, which may also increase the risk for oncogenesis in the edited cells[19]. Therefore, assessment and elimination of adverse editing events in vivo are important for the clinical translation of CRISPR-Cas9, and a practical and effective solution is highly demanded to reduce chromatin

[1]The MOE Key Laboratory of Cell Proliferation and Differentiation, School of Life Sciences, Center for Life Sciences, Genome Editing Research Center, Peking University, Beijing 100871, China. [2]Institute of Neuroscience, State Key Laboratory of Neuroscience, Key Laboratory of Primate Neurobiology, CAS Center for Excellence in Brain Science and Intelligence Technology, Shanghai Research Center for Brain Science and Brain-Inspired Intelligence, Shanghai Institutes for Biological Sciences, Chinese Academy of Sciences, Shanghai, China. [3]Department of Ophthalmology and Vision Science, Shanghai Eye, Ear, Nose and Throat Hospital, Fudan University, Shanghai, China. [4]Department of Ophthalmology, the Affiliated Hospital of Guizhou Medical University, Guiyang, China. [5]These authors contributed equally: Jianhang Yin, Kailun Fang, Yanxia Gao. ✉e-mail: huiyang@ion.ac.cn; hujz@pku.edu.cn

structural variations and exogenous DNA integration during in vivo genome editing.

Age-related macular degeneration (AMD) is a main cause of blindness and about two-thirds of the aged population over 80 years old show signs of AMD. Anti-vascular endothelial growth factor A (VEGFA) agents are the major treatment for wet AMD, including small molecules (pegaptanib) and antibodies (aflibercept)[20,21]. These strategies show unprecedented efficacy but have the risk of causing local complications due to repeated injections. In recent years, CRISPR-Cas toolboxes have been adapted to target *VEGFA* for the treatment of AMD, among which CRISPR-Cas9 has been successfully used by multiple groups to disrupt *Vegfa* in mouse models of laser-induced choroidal neovascularization (CNV)[22–25]. By subretinal injection for Cas9 and single-guide RNA (sgRNA) via ribonucleoprotein (RNP) or adeno-associated virus (AAV), the laser-induced neovascularization could be efficiently inhibited in mouse eyes[22–24].

By inhibiting repeated cleavage, Cas9TX has been reported to efficiently eliminate chromosomal translocations in edited cells ex vivo[10], but the in vivo capacity of Cas9TX to safeguard genome integrity is still lacking. Here, we applied Cas9TX to efficiently treat AMD in the mouse model. Strikingly, Cas9TX can not only greatly suppress chromosomal translocations, but also inhibit AAV integrations at the target site. We conclude that Cas9TX is a superior choice for in vivo gene-editing instead of Cas9 for in vivo gene editing.

## Results

### Chromatin structural variations arising in gene-edited murine

To disrupt *Vegfa* in the mouse eye, we adapted the split-Cas9 system by packing the long Cas9 gene into two split-intein AAV vectors as previously described[26]. We prepared one vector encoding the elongation factor-1 short (EFS) promoter-driven *Sp*Cas9[1–573]-intein and U6 promoter-driven sgRNA targeting *Vegfa* (hereafter referred to as Cas9-Nter). The other vector was prepared to encode EFS-driven intein-*Sp*Cas9[574–1368] (hereafter referred to as Cas9-Cter). We applied target sequencing analysis to evaluate 6 sgRNAs targeting exon 2 of *Vegfa* and 7 sgRNAs targeting exon 3 of *Vegfa* in the mouse neuroblast N2a cell line (Supplementary Fig 1a). CRISPResso analysis showed that the percentages of indels ranged from 4.5% to 91.5% for these sgRNAs (Supplementary Fig 1a). Accordingly, we chose two adjacent sgRNAs (sg1 and sg2) targeting exon 3 of *Vegfa* with the highest editing efficiency and used them separately for gene disruption in vivo. Note that these two sgRNAs can also target human *VEGFA*, which are potential candidates for therapeutic gene editing. 1µl of dual AAVs with split *Sp*Cas9 and *Vegfa*-sgRNA were co-injected into mouse retina at a dose of 3E9 vector genomes (vg)/eye. The editing efficiency and chromatin structural variations were monitored by PEM-seq analysis at 2-, 4-, and 12- weeks after the resulting dual AAVs were administered into the mouse eyes via subretinal injection (Fig. 1a).

To prepare PEM-seq libraries after *Vegfa* editing, we designed a shared bait primer 60 bp upstream of the proximal site (*Vegfa*-sg1, GACCCTGGTGGACATCTTCCAGG) and 69 bp upstream of the distal site (*Vegfa*-sg2, CTCCTGGAAGATGTCCACCAGGG). The editing efficiency of Cas9:*Vegfa*-sg1 was approximately 7.6%, 9.0%, and 12.7% for 2-, 4-, and, 12-weeks post AAV injection, respectively (Fig. 1b). Meanwhile, we identified 5 off-target (OT) sites for Cas9:*Vegfa*-sg1 located within *Cd96*, *Cddc85c*, *Egf1am*, *Grm4*, and *Gm39027* genes (Fig. 1c). The sequences of these off-target sites are similar to the on-target sites, bearing 2–4 nucleotide mismatches in the spacer region (Fig. 1c). We PCR-amplified the regions spanning the identified off-target sites and the amplified DNA fragments can be readily cleaved by Cas9:*Vegfa*-sg1 after 120-min incubation in vitro (Supplementary Fig. 1b), validating that they are real off-target sites for Cas9:*Vegfa*-sg1. Simultaneous cleavage induced by CRISPR-Cas9 at off- and on-target sites can generate off-target translocations[10,27]. The total percentages of off-target translocations at the *Vegfa*-sg1 site in edited cells were approximately

0.07% for 2-weeks post-injection and reached about 0.43% after four weeks (Fig. 1d). The off-target translocations retained a high level at approximately 0.21% for 12-weeks post-injection (Fig. 1d), indicating the persistence of chromosomal translocations after *Vegfa* editing. Furthermore, we confirmed the translocation products between OT1 and the target site by PCR amplification (Supplementary Fig. 1c). In addition to off-target translocations, genome-wide occasional double-stranded breaks (DSBs) may also fuse with on-target DSBs to form general translocations[10,27,28]. The percentages of general translocations at the *Vegfa*-sg1 site fell in a range from 0.21% to 0.67% at 2-, 4-, 12-weeks post-AAV injection (Fig. 1d). We also observed a high level of AAV integrations at the *Vegfa*-sg1 locus. The percentages of AAV integrations in edited cells were 28.6%, 36.4%, and 29.6% for 2-, 4-, and, 12-weeks post-AAV injection, respectively (Fig. 1e).

A similar level of chromosomal translocations was identified for the *Vegfa*-sg2 site. The Cas9:*Vegfa*-sg2 achieved indels at a frequency ranging from 7.6% to 14.1% at 2-, 4-, 12-weeks post AAV injection (Supplementary Fig. 1d). The percentages of off-target translocations were from 0.03% to 0.06%, lower than those of the *Vegfa*-sg1 site (Supplementary Fig. 1e and f), probably due to fewer off-target sites. General translocations were from 0.20% to 0.60% and AAV integrations were from 22.5% to 46.8%, similar to those of the *Vegfa*-sg1 site (Supplementary Fig. 1g and h). These results suggest that chromosomal translocations and AAV integrations universally occur during in vivo genome editing. Conversely, large deletions were barely detected after *Vegfa* editing in vivo (Supplementary Fig. 1i).

### Cas9TX achieves the treatment of AMD in vivo

Since Cas9TX can greatly reduce chromosomal structural variations during genome editing ex vivo[10], we applied Cas9TX in comparison with Cas9 for the treatment of AMD in the mouse model. The size of the optimized TREX2 fused to Cas9 was only 236 aa, so Cas9TX was supposed to be compatible with the dual-AAV package systems. In this context, we prepared a vector encoding intein-*Sp*Cas9[574–1368]-TREX2 (R163A, R165A, and R167A) (hereafter referred to as Cas9TX-Cter) to pair with the Cas9-Nter vector. To comprehensively compare the effectiveness of Cas9TX and Cas9, we employed a widely-used mouse AMD model. In general, the Bruch's membrane of normal mice was ruptured by laser, leading to the neovascularization from the choroid into subretina, in order to mimic the main characteristics of human AMD[29,30]. We induced CNV in the mouse eyes by laser treatment at 42-days post AAV injection and measured the area of CNV and the levels of VEGFA protein one week later (Fig. 2a).

The AAV copy numbers for Cas9 and Cas9TX were comparable in infected mouse eyes revealed by real-time PCR (RT-PCR), ruling out the difference in virus quality for in vivo genome editing (Supplementary Fig. 2a and b). In this context, the editing efficiency of Cas9TX was slightly higher, though not significant (*p* value=0.11), than that of Cas9 at the *Vegfa*-sg1 locus (19.4% vs 10.1%) (Fig. 2b). In regards to off-target activity, Cas9 and Cas9TX induced similar levels of indels at most off-targets, with OT2 of *Vegfa*-sg1 showing less cleavage by Cas9TX in comparison to Cas9 (1.0% vs 7.4%) (Supplementary Figs 3a and b). Moreover, Cas9TX tended to induce more small deletions at the *Vegfa* gene in comparison to Cas9 in vivo (Fig. 2c and Supplementary fig. 2c). Consequently, Cas9TX and Cas9 reduced the concentration of VEGFA protein in the whole eye to 16.9 and 16.7 pg/mL (*n* = 8), respectively, in comparison to 22.5 pg/mL (*n* = 16) of untreated wild-type (WT) group (Fig. 2d). Conversely, injection of PBS (23.4 pg/mL, *n* = 8) or injection of a single-AAV encoding Cas9-Cter (21.7 pg/mL, *n* = 8) showed no impact on the concentration of VEGFA protein (Fig. 2d). Next, we examined laser-induced CNV in mouse eyes by fluorescein fundus angiography (FFA). In comparison to Cas9-Cter-treated sample, the neovascularization was efficiently inhibited after injection of dual-AAVs expressing Cas9TX or Cas9 targeting *Vegfa*-sg1 (Fig. 2e). FFA grade was dropped from 3.3 to 2.0 or 2.3 after Cas9TX or Cas9 treatment, respectively

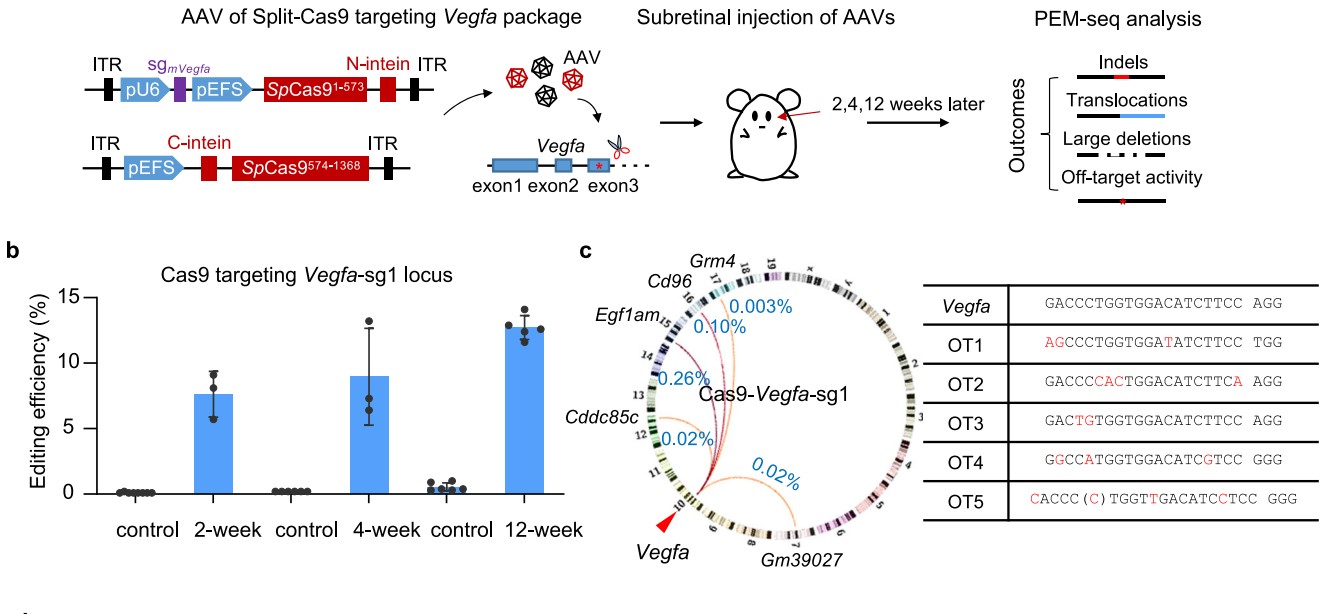

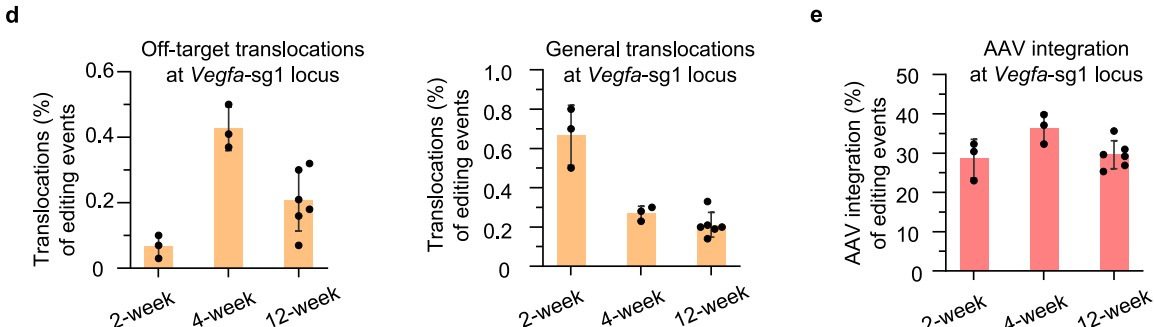

**Fig. 1 | Chromatin structural variations during in vivo genome editing.**
**a** Schematics for assessing chromosomal structural variations induced by dual-AAV-delivered *Sp*Cas9 in the mouse eye. **b** Editing efficiency for *Vegfa*-sg1 in mouse eye at 2-, 4- and 12-weeks post-transfection detected by PEM-seq. Error bars, mean ± SD (For 2-, 4-, and 12-weeks post-PBS injection mice, *n* = 8, 6, 6, respectively; for 2-, 4-, and 12-weeks post-AAV injection mice, *n* = 3, 3, 5, respectively). **c** Circos plot indicating translocations among on-target and off-targets of *Vegfa*-sg1 in mouse eye detected by PEM-seq. DNA sequences for on-target and off-targets of

*Vegfa*-sg1 are shown in the table. Mismatches of DNA sequences between on-target and off-targets are in red. **d** Percentages for off-target translocations (left) and general translocations (right) in mouse eye cloned from *Vegfa*-sg1 at 2-, 4- and 12-weeks post-transfection detected by PEM-seq. Error bars, mean ± SD (For 2-, 4-, and 12-weeks post-AAV injection mice, *n* = 3, 3, 6, respectively). **e** Percentages of AAV integrations at *Vegfa*-sg2 normalized to editing events at 2-, 4- and 12-weeks post-transfection detected by PEM-seq. Error bars, mean ± SD (For 2-, 4-, and 12-weeks post-AAV injection mice, *n* = 3, 3, 6, respectively).

(Fig. 2f). These results demonstrate that Cas9TX can efficiently treat AMD in our mouse model.

## Cas9TX almost eliminates chromosomal translocations during AMD treatment

To assess genome integrity during AMD treatment by Cas9TX or Cas9 editing, genomic DNA was extracted from mouse eyes one-week post induction of CNV for chromatin structural variation analysis by PEM-seq. The five off-target sites of *Vegfa*-sg1 were also detected in the AMD model seven weeks after CRISPR-Cas9 treatment (Fig. 3a). In contrast, only the strongest OT1 site was identified as a weak translocation hotspot in the CRISPR-Cas9TX-treated mice, however, the translocation intensity was decreased from 0.26% of Cas9 to 0.002% of Cas9TX (Fig. 3a and b). While the other four off-target translocation hotspots were barely detected in the AMD model after CRISPR-Cas9TX treatment (Fig. 3a and b). Moreover, Cas9TX also greatly reduced general translocations to 0.09% from 0.75% of Cas9 at the *Vegfa*-sg1 site in mouse eyes (Fig. 3c), further indicating the effectiveness of Cas9TX to suppress chromosomal translocations. Similar findings were obtained for the *Vegfa*-sg2 locus. Compared the Cas9TX treatment to Cas9 treatment, changes at 30-fold and 4.3-fold were detected for off-target

translocations (0.1% vs 0.003%) and general translocations (0.56% vs 0.13%), respectively, at the *Vegfa*-sg2 locus (Fig. 3d–f). Collectively, these data show that Cas9TX can be used to effectively eliminate chromosomal translocations for in vivo gene editing.

## Cas9TX suppresses AAV integration during AMD treatment

The gene-editing target site is the main locus for the integration of exogenous DNA, including AAV[13]. Since PEM-seq can capture all the editing outcomes within the editing sites under the bait primer, we extracted the sequencing junctions harboring the AAV fragments (Fig. 4a). The AAV integrations can be categorized into two types: a short AAV fragment buried in the target site or a fusion between AAV and the target site (Fig. 4a and b). We combined these two types of AAV integration products for further analysis as previously described[31,32]. The percentages of AAV integrations in the Cas9-treated AMD mice were approximately 11.0% and 13.3% at the *Vegfa*-sg1 and *Vegfa*-sg2 target sites, respectively, when normalized to total sequenced events (Fig. 4c). This means that about 52.0% and 48.5% of edited cells will have AAV integrations at the *Vegfa*-sg1 and *Vegfa*-sg2 target sites, respectively, when normalized to indels. The integration junctions were mainly enriched at the AAV inverted terminal repeat (ITR) region as previously

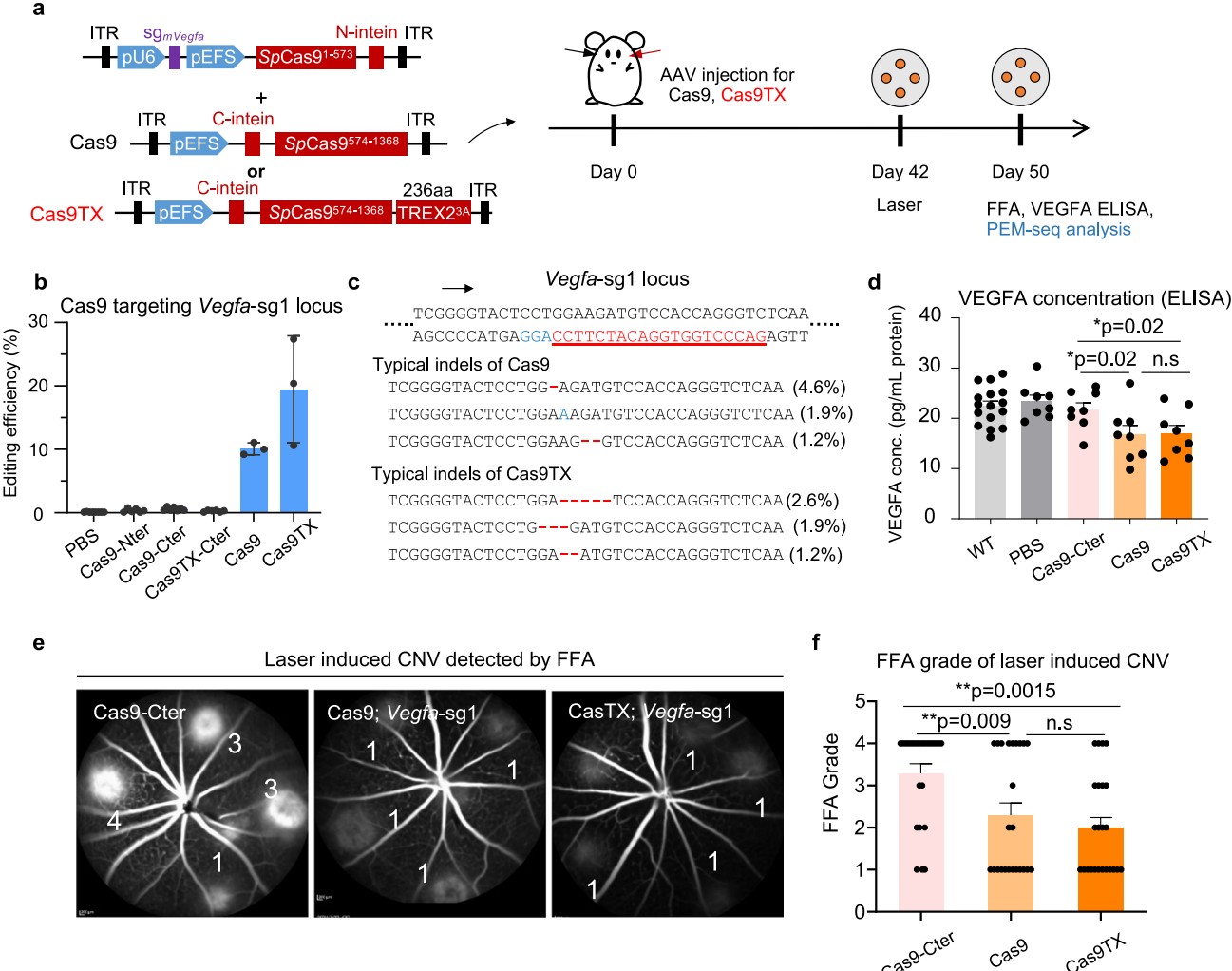

**Fig. 2 | Cas9TX is able to efficiently inhibit CNV. a** Schematics for treatment for AMD in mouse models by Cas9TX and Cas9. Cas9TX and Cas9 are packaged into dual-AAV systems. At 42-days post infection of the resulting AAVs into the mouse eyes, neovascularization is induced into the mouse eyes by laser. One week later, the neovascularization in the mouse eyes is monitored by FFA, the protein level of VEGFA is detected by ELISA, and chromosomal structural variations are detected by PEM-seq analysis. **b** Editing efficiency at *Vegfa*-sg1 for the indicated groups in mouse models of AMD detected by PEM-seq. Error bars, mean ± SD (For PBS-, Cas9-Nter-, Cas9-Cter-, Cas9TX-Cter-, Cas9-, and Cas9TX-treated mouse models, $n = 5, 5, 4, 5, 3$, and 3, respectively). **c** Examples of DNA sequences for typical indels induced by Cas9 and Cas9TX at *Vegfa*-sg1. The frequency of the products is indicated. The DNA sequence of *Vegfa*-sg1 is in red and underlined. **d** The concentration of VEGFA protein in mouse models of AMD for the indicated groups detected by ELISA. Error bars, mean ± SD (For untreated mice, $n = 16$; for PBS-, Cas9-Cter-, Cas9-, Cas9TX-treated mouse models, $n = 8$). Two-tailed *t*-test, *$p < 0.05$, n.s means no significance. **e** Measurement for neovascularization in Cas9-Cter-, Cas9-, and Cas9TX-treated mouse models of AMD by FFA at 1-week after laser induction. **f** FFA grade for Cas9-Cter-, Cas9-, and Cas9TX-treated mouse models of AMD at 1-week after laser induction. Mean ± SD from 24 replicates for each group. Two-tailed *t*-test, **$p < 0.01$, n.s means no significance.

described[13] (Fig.4d). In contrast to Cas9 injection, the percentages of AAV integrations at the *Vegfa*-sg1 site were only about 0.01% for injection of PBS, 0.14% for injection of AAV encoding Cas9-Nter, and 0.16% for injection of AAV encoding Cas9-Cter (Fig.4e). Strikingly, Cas9TX significantly reduced the levels of AAV integrations to 0.73% at the *Vegfa*-sg1 site, but higher than 0.06% for injection of Cas9TX-Cter (Fig. 4c). Similarly, the percentages of AAV integrations in the *Vegfa*-sg2 site dropped down to 1.51% for injection of dual-AAV encoding Cas9TX (Supplementary fig. 4a and b). Therefore, AAV integrations may pose a great threat to gene-edited cells in vivo and Cas9TX can largely alleviate this type of deleterious editing byproducts.

## Discussion
AAV-mediated CRISPR-Cas shows great potential for the treatment of eye diseases, including AMD and Leber congenital amaurosis type 10 (LCA10)[1,22–24]. Thereinto, the Cas9 therapy to LCA10 is in the phase1/2 clinical trial (NCT03872479). However, the generation of

chromosomal translocations and AAV integrations induced by Cas9 has not been explored in these diseases. Cas9 can induce chromosomal translocations at a frequency of approximately 1- 3% in cultured human primary T cells and mouse embryonic stem cells[10,33], and the translocation levels were further elevated to as high as 1–10% in edited 293 T cells[27]. Here, we find that chromosomal translocations still occur at a frequency of approximately 1% in CRISPR-Cas9-edited mouse eyes and the translocations can persist in vivo for more than 12 weeks (Fig. 1d). Though the translocation levels for in vivo editing are relatively lower than those arising in ex vivo editing, the unpredictable chromosomal translocation is still a potential risk for gene editing in clinical. Moreover, unlike malignant tumors, diseases like AMD have a severe impact on the quality of patients' life but are not fatal. In this regard, any risk potentially leading to cancer should be avoided during gene-editing therapy. In this study, Cas9TX can be delivered to mice via a split-AAV system and outperforms Cas9 in all the tested aspects. Strikingly, the translocation levels show hundreds of fold differences

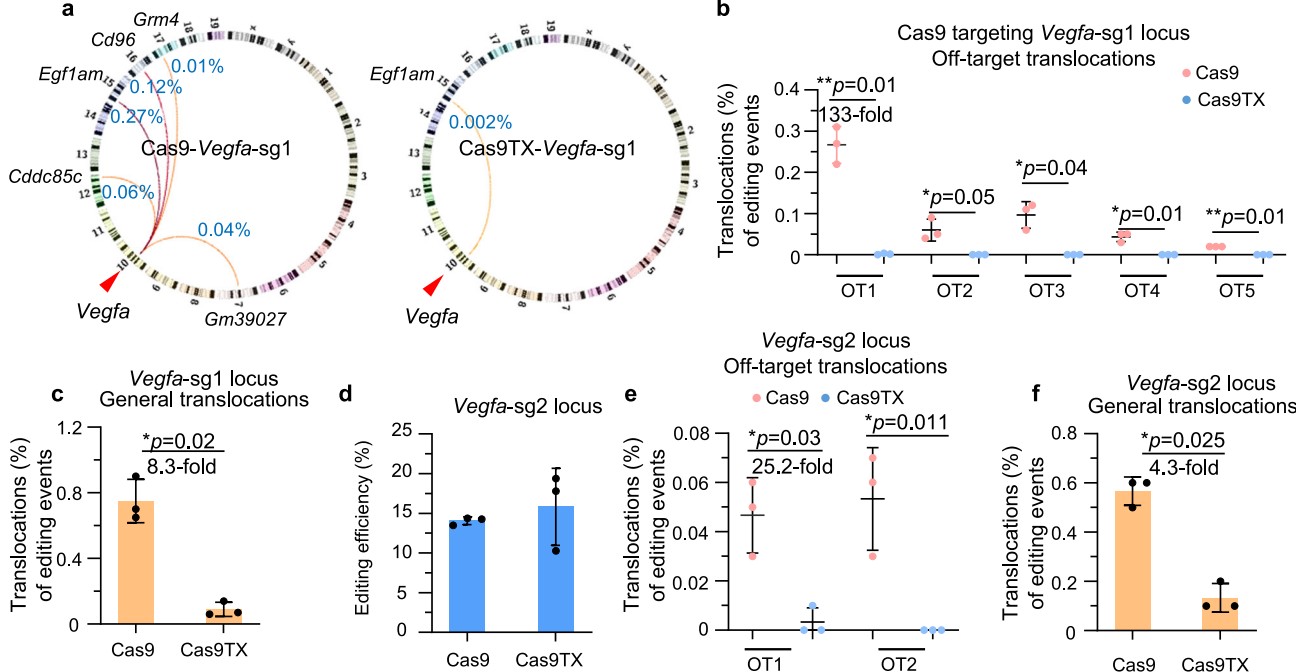

**Fig. 3 | Cas9TX nearly eliminates chromosomal translocations in vivo. a** Circos plot indicating translocations among on-target and off-targets of *Vegfa*-sg1 in Cas9- and Cas9TX-treated mouse models of AMD detected by PEM-seq. **b** and **c**. Percentages for off-target translocations (**b**) and general translocations (**c**) cloned from *Vegfa*-sg1 in Cas9- and Cas9TX-treated mouse models of AMD detected by PEM-seq. Mean ± SD from three biological replicates. Two-tailed *t*-test, *$p < 0.05$,

**$p < 0.01$. d** Editing efficiency at *Vegfa*-sg2 in Cas9- and Cas9TX-treated mouse models of AMD detected by PEM-seq. Mean ± SD from three biological replicates. **e** and **f** Percentages for off-target translocations (**e**) and general translocations (**f**) cloned from *Vegfa*-sg2 in Cas9- and Cas9TX-treated mouse models of AMD detected by PEM-seq. Mean ± SD from three biological replicates. Two-tailed *t*-test, *$p < 0.05$, **$p < 0.01$.

between the original Cas9 and Cas9TX during the treatment of AMD in the mouse model, indicating a potentially better performance of Cas9TX to suppress chromosomal translocation in vivo.

The integration of virus DNA into the genome threatens genome integrity and is associated with perturbed gene expression or even human malignancies[19,34]. As the most currently used gene delivery tool in clinical trials, AAV renders gene-edited cells to suffer from high levels of virus-fragment integrations during CRISPR-Cas9 editing in vivo (~50% in edited cells in this study)[13,14]. A long-term study of AAV-delivery gene therapy in dogs showed that the AAV payload can be integrated into the host's genome close to genes that control cell growth, which leads to unexpected clone expansion[12]. To our knowledge, Cas9TX is currently the only reported editing tool to reduce AAV integration. Since microhomology is important for vector integration[19,35], Cas9TX may reduce the levels of AAV integrations by processing the broken ends to avoid the formation of microhomology as well as shortening the time of DSB exposure.

Different from chromosomal translocations and AAV integrations, large deletions are much less frequently observed during in vivo CRISPR-Cas9 editing in mouse eyes (Supplementary fig. 1g), which may need to be further validated in other target sites, tissues, or species. Moreover, Cas9TX can be delivered into cells via electroporation, RNP, and split-AAV as Cas9[10]. Therefore, Cas9TX should be compatible with almost all the gene-editing scenarios suitable for Cas9 with better safety at a comparable editing ability.

## Methods
### Vector and gRNA sequences
The vector information is provided in Source data. g*Vegfa*-1: 5'-gaccctggtggacatcttccagg-3'; sg*Vegfa*-2: 5'-ctcctggaagatgtccaccaggg-3'. Other tested sgRNAs targeting Vegfa are listed in Supplementary Fig. 1a.

### Animals
All animal experiments were conducted in compliance with the Guide for the Laboratory Animal Administration Regulation (2017) issued by the National Science and Technology Committee (China) and the laboratory animal administration regulations (Shanghai, Jiangsu). The care and use of animals were reviewed and approved by the Institutional Animal Care and Use Committee (IACUC) of HuiGene Therapeutics Co., Ltd. Mouse conditions were monitored by professional institutional staff. Adult (6–8 week old) male and female SPF C57BL/6 J mice (half and half) were used in the study. Animals were housed socially in standard cages with a feeding box and a water bottle. The environment of the laboratory animal facilities and cages used were in accordance with China National Standard: GB14925-2010 – Laboratory Animals – Requirements of Environment and Housing Facilities (2010-12-23). The rooms were controlled and monitored for humidity (targeted range 40–70%, actual range 20.6–77.5%) and temperature (targeted range 18 °C–26 °C, actual range 17.6 °C–25.1 °C) with 10 to 20 air changes/hour. The room was on a 12-hour light/dark cycle except when interruptions were required by study activities. The rooms in which the animals were housed have been documented in the facility records. Euthanasia was performed by appropriately trained persons and mice underwent euthanasia by $CO_2$ exposure.

### AAV package and preparation
Viral particles of AAV9 were packaged in co-transfected HEK293T cells with the other two plasmids: pAAV-Rep-Cap and pAAV-Helper. After harvest, viral particles were purified with a heparin column (GE HEALTHCARE BIOSCIENCES) and then concentrated with an Ultra-4 centrifugal filter unit (Amicon, 100,000 molecular weight cutoff). Titers of viral particles were determined by qPCR and diluted to 3E12 particles/ml for stock.

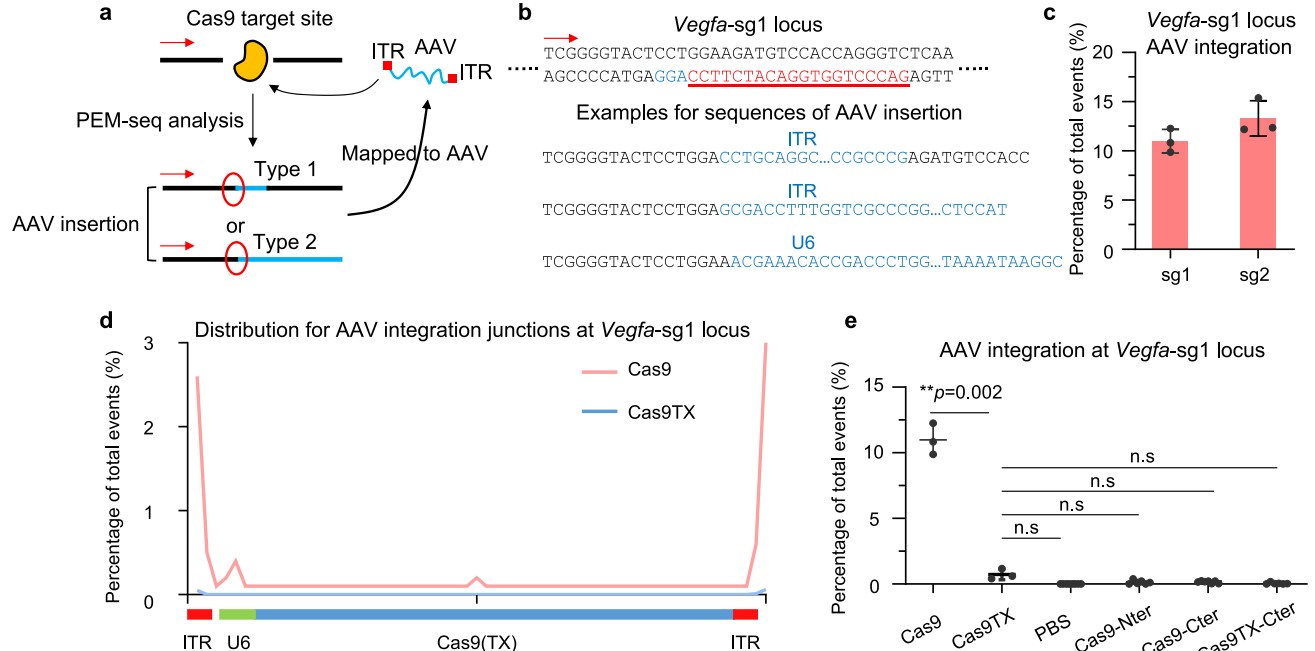

**Fig. 4 | Cas9TX reduces AAV integrations to the background levels in vivo.**
**a** Schematics for the identification for AAV integrations by PEM-seq. Red arrows indicate the primers placed on the *Vegfa* locus in the genome. AAV junctions can be divided into two types as indicated. **b** Examples of DNA sequences for AAV integrations at *Vegfa-sg1*. The blue letters indicate the integrated AAV sequence. **c** Frequency for AAV integrations at *Vegfa*-sg1 and *Vegfa*-sg2 normalized to total

sequence events detected by PEM-seq. Mean ± SD from 3 replicates for each group. **d** Distribution for AAV integration junctions at the AAV vectors. **e** Frequency for AAV integrations at *Vegfa*-sg1 for the indicated groups in mouse models of AMD detected by PEM-seq. Error bars, mean ± SD (For Cas9-, Cas9TX-, PBS, Cas9-Nter-, Cas9-Cter-, and Cas9TX-Cter-treated mouse models, *n* = 3, 3, 7, 6, 6, and 6, respectively). Two-tailed *t*-test, *p < 0.05, n.s means no significance.

## Subretinal injection

8-week-old C57BL/6 mice were used for subretinal injection. Pupils were first dilated with 0.5% tropicamide and 0.5% phenylephrine eye drops (Santen Pharmaceutical). Mice were then anesthetized with a mixture of zoletil (60 mg/kg) and xylazine (10 mg/kg). A small hole at the slight posterior to the limbus was punctured with a sterile 31 G 1/2 needle. One microliter of AAV virus and vehicle was subretinally injected slowly through the hole using a Hamilton syringe with a 33 G blunt needle. The working dose of AAV in the retina is 3E9 vg/eye. Eyes with severe hemorrhage or damage were excluded from further study.

## Laser-induced CNV mouse models

4 weeks after AAV injection, mice were used for CNV induction. Briefly, pupils were dilated with tropicamide eye drops (Santen Pharmaceutical), and mice were anesthetized with a mixture of zoletil (60 mg/kg) and xylazine (10 mg/kg). Laser photocoagulation was performed using NOVUS Spectra (LUMENIS). The laser parameters used in this study were: 532 nm wavelength, 50 ms exposure time, 200 mW power, and 50 μm spot size. 3 to 4 laser burns around the optic disc were induced for FFA detection. 20 laser burns around the optic disc were induced for ELISA detection. Mice with vitreous hemorrhage were excluded from the study. CNV and ELISA analysis was conducted 8 days after laser burn.

## Fundus fluorescein angiography (FFA) analysis in mice

Days after CNV induction, mice were subjected to fundus fluorescein angiography to analyze the growth of CNV. Pupils were dilated with tropicamide eye drops, then mice were anesthetized with a mixture of zoletil (62.5 mg/kg) and xylazine (10 mg/kg). Fluorescein Sodium (150 mg/kg) was quickly administered by tail intravenous injection. FFA images were obtained by an ophthalmic laser diagnosis device (Heidelberg).

## DNA and protein purification from the retina

DNA of mouse retina and RPE was extracted with DNeasy Blood & Tissue Kit (Qiagen). The protein of the mouse eye was extracted with RIPA lysis buffer.

## ELISA for VEGFA protein quantitation

The whole eye was collected for ELISA. To perform VEGFA ELISA, 20 laser burns were induced in each eye 6 weeks after AAV injection. Eyes were enucleated 8 days post-induction and lysed with RIPA lysis buffer. VEGFA protein levels were determined using a Quantikine ELISA kit (MMV00, R&D SYSTEMS) according to the standard protocol.

## PEM-seq analysis for translocations and AAV integrations

The operation of PEM-seq was described previously[27,36]. In general, biotinylated primer 5'-CCCCTCCTTGTACCACTGTCCTCCTG-3' was set at about 200-bp from the Cas9 target site for primer extension. Extension products were enriched by Streptavidin beads (Thermo Fisher, 65001) followed by bridge adapter ligation. Next, locus-specific primer CGTTACAGCAGCCTGCACAGC was used for nested PCR followed by Illumina Hiseq sequencing.

Translocation hotspots with a sequence similar to the target site (≤8 nt mismatches for both sgRNA and PAM sequences) and with junctions at the presumable *Sp*Cas9 cut-site were considered as off-target sites. General translocations were calculated by excluding junctions ± 20 kb around the target sites and ± 100 bp around the off-target sites.

There are two main types of AAV integrations in the text. One is that the entire inserted fragments can be aligned to the AAV backbone. The other is fused with the target site, which still has the potential to be large vector integrations. Therefore, we remapped these reads to the genome and then the AAV backbone to find AAV integrations. We used bwa-mem to do the alignment with a default seed length of 20 bp.

Notably, when calculating the editing efficiency and translocations, the editing events refer to indels and translocations but don't

contain AAV integrations. When calculating the AAV integrations, the editing events refer to indels, translocations, and AAV integrations. The total events refer to indels, translocations, uncut, and AAV integrations.

## Targeting sequencing analysis for off-target indels

As for *Vegfa*-sg1, primers GCAACTCCAACAACTCAATTTG and GCTCCAACAGGGAGCTTACAC were used for OT1 amplification. GAGTGGGTTTACAGGGTCTGG and GTCCTCAAAGCCTGGGAGCTG were used for OT2 amplification. GGCCCCTCTCATGAGAAGAGC and GGCAGAATCAGGCTGGGTGG were used for OT3 amplification. CACGGCTCCCGACTTGAGTG and GAGAACCCAGCAGCAGTGGG were used for OT4 amplification. GATAAGTGAGTTACCCATAGCC and GCCTGACACACACATACACCC were used for OT5 amplification. As for *Vegfa*-sg2, primers CAACTCAATTTGAAGACGGG and GAGCTTACA-CAGAGTAGCTTG were used for OT1 amplification. GGAGACTATA-GACTCCACCTC and GTGTGGTTGCTCTCCAGCCTG were used for OT2 amplification. Amplicons underwent Illumina Hiseq sequencing followed by CRISPResson (version 1.0.8) analysis with default parameters.

## PCR amplification for translocation products

Primers 5′-CACACAGGACGGCTTGAAGATG-3′ and 5′-CTTGCA-CAAAGCCCTAGTTTTG-3′ were used for amplifying the translocations between the on-target site and OT1 (35 cycles). The PCR products were ligated into a plasmid and transfected into *Escherichia coli* followed by Sanger sequencing for the single clone.

## Statistics and reproducibility

Statistics were performed using a two-tailed Student's *t*-test. Data were presented as the mean ± SD for more than 3 biological repeats, and $p < 0.05$ was considered significant.

## Reporting summary

Further information on research design is available in the Nature Portfolio Reporting Summary linked to this article.

## Data availability

Original PEM-seq data for mouse eye have been deposited into the NCBI Gene Expression Omnibus (GEO) database under accession code GSE218818. Source data are provided in this paper. Source data are provided with this paper.

## Code availability

Code for PEM-seq analysis is available: https://github.com/liumz93/PEM-Q.

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

## Acknowledgements

We thank the Flow Cytometry Core at National Center for Protein Sciences at Peking University for technical support. This work was supported by the National Key R&D Program of China (2022YFC3400201), the Ministry of Agriculture and Rural Affairs of China, the NSFC (32122018 to J.Hu, 31925016 to H.Y., 82171102 to J.Ho.), the Clinical Medicine Plus X–Young Scholars Project (No. PKU2020LCXQ021), Basic Frontier Scientific Research Program of Chinese Academy of Sciences From 0 to 1 original innovation project (ZDBS-LY-SM001), Shanghai Municipal Science and Technology Major Project (2018SHZDZX05), Project of Shanghai Municipal Science and Technology Commission (20MC1920400), and the PKU-TSU Center for Life Sciences.

## Author contributions

J.Y., K.F., Y.G., J.Ho., H.Y., and J.Hu. designed the experiments; J.Y., S.Y., L.O., and C.X. performed the PEM-seq analysis; K.F., Y.G., W-WW., and WW. constructed the plasmids and performed the AMD-associated experiments in mouse; J.Y., K.F., H.Y., and J.Hu. analyzed the data; J.Y., K.F., J.Ho., H.Y., and J.Hu. wrote the paper.

## Competing interests

The authors declare no competing interests.
