## [Peer Review File · Nature Communications]

Safeguarding genome integrity during gene-editing therapy of age-related macular degenerationREVIEWER COMMENTS

Reviewer #1 (Remarks to the Author):

The authors describe the in vivo application of an engineered Cas9 variant previously demonstrated to decrease chromosomal translocation via inhibition of repeat cleavage. In this work, the authors demonstrate comparable rescue to wildtype cas9 in a laser-induced model of neovascularization when targeting VEGFA. Furthermore, authors show an overall decrease in chromosomal translocations during treatment. Surprisingly, authors also observe a substantial decrease in viral vector genomic integration into host genome. This work is highly valuable and highlights the successful in vivo application of a recently developed Cas9 variant which can be used to improve the safety and translatability of CRISPR/Cas9 for clinical applications in the future. Authors observed unexpectedly high AAV genome integration using SpCas9 (Figure 1E). Meanwhile, an exploration of the mechanism related to the variant's impact on AAV genome insertion would be highly useful to solidify these statistically significant but relatively slight decreases in integration events. The methodologies provided are thorough and reproducible.

Reviewer #2 (Remarks to the Author):

The manuscript by Hu and colleagues describes the analysis of genome editing and phenotypic outcomes in the mouse retina following injection of dual AAV vectors delivering SpCas9 or a SpCas9-Trex2 (SpCas9TX) fusion targeting the Vegfa locus. Genome editing outcomes are evaluated for two different sgRNAs using PEM-seq, which provides an assessment of indel rates and the presence of large deletions, AAV insertions or chromosomal translocations. PEM-seq assessments indicate that SpCas9TX achieves similar rates of genome editing at the target site while reducing the rate of translocations with off-target sites as well as the rate of AAV integration at the target site. In addition, SpCas9TX treatment achieves similar rates of suppression of Laser-induced choroidal neovascularization (CNV) relative to SpCas9, and similar suppression of VEGFA protein levels.

Overall, this is a focused paper with important validation for in vivo editing outcomes for SpCas9TX that is consistent with the authors observations for SpCas9TX editing outcomes in cell lines in their original publication (reference 10). There are a few aspects of the study that require more detailed characterization, but once addressed this will be an impactful manuscript that may convince many laboratories to consider SpCas9TX as their nuclease of choice for in vivo editing when using AAV delivery.

Important questions to address:

1) What are the editing (indel) rates at the individual off-target sites for Vegfa-sg1 & -sg2? This question should be addressed by targeted amplicon sequencing. In particular, it will be important to confirm that the off-target editing rates are not higher for the SpCas9TX nuclease relative to SpCas9. Since SpCas9TX presumably decreases precise repair at a cleaved locus, it would not be surprising to find that the off-target editing rates increase for this variant.

2) The description of the quantification of AAV integrations is confusing throughout the manuscript. Assessing the rates of AAV integration is a critical component of the study, so it is important to have clarity about this observation in the text. For example on line 107

"The percentages of AAV integrations in edited cells were 28.6%, 36.4%, and 29.6% for 2-, 4-, and, 12-weeks post AAV injection, respectively (Fig. 1e)."

The reported percentages are higher than the reported indel rates for the target site (12.7%). Is it really the case that AAV integration rates are higher than indel rates? If so these disruptive insertions,

which are higher for SpCas9 than SpCas9TX should yield a greater impact for SpCas9 on VEGFA production than SpCas9TX.

3) The authors rely on PEM-seq for the identification of off-target site edited by SpCas9, but this is predicated on the presence of translocations between the target sequence and an off-target site. Some active off-target sites may have very inefficient translocations with the target site due to sequence or cellular biases. It would be useful to evaluate the off-target profile for editing at the Vegfa locus in cell culture where the editing rates at the target and any off-target sites should be much higher. The cell culture off-target analysis (and translocation rates) for Cas9 and Cas9TX could then be used as a proxy for sites to focus evaluation of the in vivo editing outcomes.

Minor questions/comments:

4) Supplementary fig 2 is confusing. It appears that there are six different treatment groups that are being evaluated, with some of the experimental groups containing only one of the two AAV vectors. However, this is not explained in the text or the Figure 3 schematic of the experiment, which only indicates two different treatment groups. The experimental layout needs to be clarified to facilitate the interpretation of the data that is presented. The figure legends (Fig 3 and SF2) should also indicate the size of the cohort for each treatment group.

5) The authors indicate in the text that "Cas9TX tended to induce more small deletions to disrupt the Vegfa gene in comparison to Cas9 in vivo". While this is true, the key question is what fraction of the indel (insertions or deletions) are out-of-frame alleles. If the Cas9 insertions are primarily +1, as indicated from the sequencing data, they should be equally disruptive to an out of frame deletion. In addition, one of the primary deletion alleles for SpCas9TX is a 3 bp deletion, which could potentially produce be a functional gene product. The author should report the distribution of out-of-frame indels for each treatment group. Based on the data, SpCas9 may be producing more gene disruptive alleles than SpCas9TX

6) The acronym CNV is not defined in the manuscript.

REVIEWER COMMENTS

Reviewer #1 (Remarks to the Author):

The authors describe the in vivo application of an engineered Cas9 variant previously demonstrated to decrease chromosomal translocation via inhibition of repeat cleavage. In this work, the authors demonstrate comparable rescue to wildtype cas9 in a laser-induced model of neovascularization when targeting VEGFA. Furthermore, authors show an overall decrease in chromosomal translocations during treatment. Surprisingly, authors also observe a substantial decrease in viral vector genomic integration into host genome. This work is highly valuable and highlights the successful in vivo application of a recently developed Cas9 variant which can be used to improve the safety and translatability of CRISPR/Cas9 for clinical applications in the future. Authors observed unexpectedly high AAV genome integration using SpCas9 (Figure 1E). Meanwhile, an exploration of the mechanism related to the variant's impact on AAV genome insertion would be highly useful to solidify these statistically significant but relatively slight decreases in integration events. The methodologies provided are thorough and reproducible.

Thanks for the kind comments and thanks for the referee's time to review our work. With regards to AAV integration, we hypothesized that AAV vector is broken into fragments before genomic integration and the exonuclease domain of Cas9TX can digest these free fragments to reduce vector integrations. We fully agree with the Reviewer that vector integration is a question and needs to be further explored in the future. I wish our method can help resolve this gene-editing threat.

Reviewer #2 (Remarks to the Author):

The manuscript by Hu and colleagues describes the analysis of genome editing and phenotypic outcomes in the mouse retina following injection of dual AAV vectors delivering SpCas9 or a SpCas9-Trex2 (SpCas9TX) fusion targeting the *Vegfa* locus. Genome editing outcomes are evaluated for two different sgRNAs using PEM-seq, which provides an assessment of indel rates and the presence of large deletions, AAV insertions or chromosomal translocations. PEM-seq assessments indicate that SpCas9TX achieves similar rates of genome editing at the target site while reducing the rate of translocations with off-target sites as well as the rate of AAV integration at the target site. In addition, SpCas9TX treatment achieves similar rates of suppression of Laser-induced choroidal neovascularization (CNV) relative to SpCas9, and similar suppression of VEGFA protein levels.

Overall, this is a focused paper with important validation for in vivo editing outcomes for SpCas9TX that is consistent with the authors' observations for SpCas9TX editing outcomes in cell lines in their original publication (reference 10). There are a few aspects of the study that require more detailed characterization, but once addressed this will be an impactful manuscript that may convince many laboratories to consider SpCas9TX as their nuclease of choice for in vivo editing when using AAV delivery.

Thanks for the referee's constructive comments and the revised manuscript has been greatly improved following the Reviewers' suggestions. A point-to-point response to the questions is below.

Important questions to address:

1) What are the editing (indel) rates at the individual off-target sites for *Vegfa*-sg1 & -sg2? This question should be addressed by targeted amplicon sequencing. In particular, it will be important to confirm that the off-target editing rates are not higher for the SpCas9TX nuclease relative to SpCas9. Since SpCas9TX presumably decreases precise repair at a cleaved locus, it would not be surprising to find that the off-target editing rates increase for this variant.

Response: Thanks for the insightful suggestions. We performed target sequencing at the on-target and off-targets for *Vegfa*-sg1 and *Vegfa*-sg2 loci in mouse eyes followed by CRISPResso analysis. In comparison to Cas9, Cas9TX showed a relatively higher (but not significant) on-target editing rate but induced a similar or lower level of indels at most tested off-targets for the two *Vegfa* loci (Rebuttal Figure 1). Given the different and more complex process for off-target cleavage by Cas9 (Bravo et al.,

2022 Nature, PMID: 35236982), we hypothesized that Cas9TX might impair the searching rate for target sequence, which more significantly affect the infrequent cleavage at off-target sites *in vivo*, in line with previous reports that the shorter duration time of Cas9 can improve editing fidelity (Shin et al., 2017 Sci Adv PMID: 28706995).

Rebuttal Figure 1. Percentage of indels at *Vegfa-sg1* (left) and *Vegfa-sg2* (right) loci for mouse eye samples detected by target sequencing. Mean \pm SD from three biological replicates, two-tailed *t*-test, n.s. means no significance, **p* < 0.05, ***p* < 0.01.

2) The description of the quantification of AAV integrations is confusing throughout the manuscript. Assessing the rates of AAV integration is a critical component of the study, so it is important to have clarity about this observation in the text. For example on line 107

“The percentages of AAV integrations in edited cells were 28.6%, 36.4%, and 29.6% for 2-, 4-, and, 12-weeks post AAV injection, respectively (Fig. 1e).”

The reported percentages are higher than the reported indel rates for the target site (12.7%). Is it really the case that AAV integration rates are higher than indel rates? If so these disruptive insertions, which are higher for SpCas9 than SpCas9TX should yield a greater impact for SpCas9 on VEGFA production than SpCas9TX.

Response: We are sorry for the inconvenience. With regards to describing AAV integrations at the Cas9 target site in Fig. 1e, we normalized the AAV integrations to the editing events (indels, translocations, and AAV integrations). We have revised the figure y axis to make it clear for readers to understand. If we normalized the AAV integrations to total events as the same normalization way for indels, the AAV integration rates are 3.0%, 5.3%, and 5.4% in all cells for 2-, 4-, and, 12-weeks post AAV injection, respectively. Note that the AAV integration rate can indeed be very high depending on experiment conditions, work done by the Gyorgy group (Hanlon et al., 2019, Nature Communications, PMID: 31570731) showed that the percentage of AAV integration in various editing scenarios *in vivo* can reach up to ~47% in their manuscript.

Both indels and AAV integration can contribute to the disruption of VEGFA production. In this regard, the percentage of indels caused by Cas9 and Cas9TX is 10.1% and 19.5% (Figure 2b) and the percentage of AAV integrations of Cas9 and Cas9TX is 11.0% and 0.7% (Figure 4e), respectively. The disruption rate of Cas9 and Cas9TX based on indels and AAV integrations are 21.1% and 20.2% at the *Vegfa*-sg1 locus. Thus, we detected a similar level decrease of VEGFA by Cas9 and Cas9TX (Figure 2d).

3) The authors rely on PEM-seq for the identification of off-target site edited by SpCas9, but this is predicated on the presence of translocations between the target sequence and an off-target site. Some active off-target sites may have very inefficient translocations with the target site due to sequence or cellular biases. It would be useful to evaluate the off-target profile for editing at the *Vegfa* locus in cell culture where the editing rates at the target and any off-target sites should be much higher. The cell culture off-target analysis (and translocation rates) for Cas9 and Cas9TX could then be used as a proxy for sites to focus evaluation of the *in vivo* editing outcomes.

Response: Thanks for the helpful suggestions. It has been accepted that the formation of chromosomal translocations is mainly dependent on the double-stranded break (DSB) frequency and the proximity of two broken ends (Alt et al., 2013 Cell, PMID: 23374339;), and we agree with the Reviewer that higher editing rates at the target site and off-target sites in culture than *in vivo* might lead to more chromosomal translocations and thus help to identify more off-targets. In this regard, we performed PEM-seq analysis for *Vegfa* locus edited by Cas9 in both mESC and N2a cell lines. Taking the *Vegfa*-sg2 locus as an example (*Vegfa*-sg1 doesn't work well in mESC (4.9% indel rate)), PEM-seq analysis showed that the editing efficiency at *Vegfa*-sg2 locus in mESC and N2a cells are much higher than *in vivo* (32.7%, 71.3% v.s. 14.1%). But the total translocation rate in mESC and N2a is much lower than that *in vivo* (0.37%, 0.47% vs 1.13%), and we didn't identify more off-targets in these two cells for *Vegfa*-sg2 locus, probably due to different behaviors of these two cell lines.

Rebuttal Figure 2. Editing efficiency (left panel) and off-target translocation (right panel) for

Vegfa locus2 edited by Cas9 detected by PEM-seq. Mean \pm SD from three biological replicates.

Minor questions/comments:

4) Supplementary fig 2 is confusing. It appears that there are six different treatment groups that are being evaluated, with some of the experimental groups containing only one of the two AAV vectors. However, this is not explained in the text or the Figure 3 schematic of the experiment, which only indicates two different treatment groups. The experimental layout needs to be clarified to facilitate the interpretation of the data that is presented. The figure legends (Fig 3 and SF2) should also indicate the size of the cohort for each treatment group.

Thanks for the kind suggestions. We excluded groups containing one of the two AAV vectors. We also added the size of the cohort for each treatment group.

5) The authors indicate in the text that “Cas9TX tended to induce more small deletions to disrupt the Vegfa gene in comparison to Cas9 *in vivo*”. While this is true, the key question is what fraction of the indel (insertions or deletions) are out-of-frame alleles. If the Cas9 insertions are primarily +1, as indicated from the sequencing data, they should be equally disruptive to an out of frame deletion. In addition, one of the primary deletion alleles for SpCas9TX is a 3 bp deletion, which could potentially produce a functional gene product. The author should report the distribution of out-of-frame indels for each treatment group. Based on the data, SpCas9 may be producing more gene disruptive alleles than SpCas9TX

Thanks for the constructive suggestions. We defined indels containing 3n (n=1, 2, 3...) bp insertions or deletions as the in-frame products and re-analyzed the PEM-seq data for Cas9- and Cas9TX-edited samples *in vivo* (Rebuttal Figure 3). We found that Cas9TX still induces more out-of-frame indels than Cas9 at the *Vegfa*-sg1 locus (13.4% v.s. 9.2%) and similar levels of out-of-frame indels at the *Vegfa*-sg2 locus (12.6% v.s. 13.4%). Cas9TX can indeed induce more in-frame indels at both *Vegfa* loci (6.0% v.s. 1.0% at *Vegfa*-sg1, 3.2% v.s. 0.7% at *Vegfa*-sg2), but it's hard to judge that whether these in-frame products generate functional VEGFA. Based on these results and to be more accurate, we changed the sentence to “Cas9TX tended to induce more small deletions at the *Vegfa* gene in comparison to Cas9 *in vivo*”

Rebuttal Figure 3. Percentage of out-of-frame and in-frame indels normalized to total events for the indicated samples.

6) The acronym CNV is not defined in the manuscript.

Thanks very much for the suggestions. We added the full name for CNV (choroidal neovascularization) in the revised manuscript.

REVIEWERS' COMMENTS

Reviewer #1 (Remarks to the Author):

Thank you for your diligent efforts. All concerns have been addressed.

Reviewer #2 (Remarks to the Author):

In their revised manuscript, the authors have addressed all of my concerns with the original document.